

# Study on variations in lidar ratios for Shanghai based on Raman lidar

Tongqiang Liu[1], Qianshan He[2,3,*], Yonghang Chen[1,*], Jie Liu[2], Qiong Liu[1], Wei Gao[2], Guan Huang[1], Wenhao Shi[1]

[1]College of Environmental Science and Engineering, Donghua University, Shanghai 201620, China
[2]Shanghai Meteorological Service, Shanghai, China
[3]Shanghai Key Laboratory of Meteorology and Health, Shanghai, China

*Correspondence to*: Qianshan He (oxeye75@163.com) and Yonghang Chen (yonghangchen@dhu.edu.cn)

**Abstract.** Accurate Lidar ratios (LR) and better understanding of their variation characteristics can not only improve the retrieval accuracy of parameters from elastic lidar, but also play an important role in assessing the impacts of aerosols on the climate. Using the observational data of Raman lidar in Shanghai from 2017 to 2019, the LR at 355 nm were retrieved and their variations and influencing factors were analyzed. Within the height range of 0.5 km–5 km, about 90% of the LR were distributed in 10 sr–80 sr with an average value of 41.0±22.5 sr, and the LR decreased with the increase of height. The volume depolarization ratios ($\delta$) were positively correlated with LR, and they also decreased with the increase of height, indicating that the vertical distribution of particle shape was one of the influencing factors of the variations of LR with height. LR had a strong dependence on the original source of the air masses. Affected by the aerosol transported from northwest of Shanghai, the average LR was the largest, 44.2±24.7 sr, accompanied by the most irregular particle shape. The vertical distributions of LR were affected by the atmospheric turbidity, with the greater gradient of LR under the clean conditions. The LR above 1km could be more than 80 sr, when Shanghai was affected by the biomass burning aerosols.

## 1 Introduction

The increase of aerosol loadings leads to the haze pollution and the decrease of atmospheric visibility (Wang et al., 2010), and also affects the earth's radiation budget and water cycle in various ways (Koren et al., 2004; Twomey, 1977). The knowledge of aerosol vertical distribution and its absorption properties are required to assess the impacts of aerosols on climate change and study the mechanism of air pollution (Ferrare et al., 2001; Sicard et al., 2011).

As an active remote sensing instrument, elastic scattering lidar can obtain the vertical distribution information of aerosols, however, it is necessary to assume the aerosol extinction to backscattering ratios (i.e., lidar ratios, LR) in the retrieval process (Fernald, 1984; Welton et al., 2001), which can result in significant errors for the extinction coefficient and followed by aerosol optical depth (AOD). To our knowledge, the LR at 355nm, 532nm and 1064nm are usually assumed to be 50 sr in China (Fan et al., 2018; Gong et al., 2015; Lv et al., 2020; Ma et al., 2019). In addition, the CALIOP (Cloud-Aerosol Lidar with Orthogonal Polarization) onboard the CALIPSO (Cloud-Aerosol Lidar Infrared Pathfinder Satellite





Observations) can observe the vertical distribution of global aerosol optical properties, and its data products have been widely used around the world (Kim et al., 2018). The CALIOP algorithm first determines the type of aerosol according to the aerosol classification algorithm, and then uses the lookup table of multiple types of aerosol to determine the LR (Kim et al., 2018; Omar et al., 2009). Therefore, the quality of the CALIOP aerosol products depends on the accuracy of aerosol type

identification and the consistency between the actual LR and that in the look-up table (Painemal et al., 2019). LR are complicated functions of time and space, which depend on the aerosol size distribution and particle composition (Reagan et al., 1988). LR are also affected by meteorological elements (Salemink et al., 1984), such as relative humidity (RH), which can change aerosol particle size distribution and refractive index (Young et al., 1993). Therefore, good knowledge of accurate LR and their variation characteristics can not only improve the retrieval accuracy of parameters from elastic lidar,

but also obtain information on aerosol types to trace the source of pollution (Franke et al., 2001).

According to definition of LR in Müller's (Müller, 2003) study:

$$LR = \frac{4\pi}{\omega P(\pi)} \qquad (1)$$

LR was negatively correlated with the phase function at 180° ( $P(\pi)$ ) and the single-scattering albedo ( $\omega$ ). The $P(\pi)$ was related to sphericity of the particle which can be obtained from the polarization lidar. The $\omega$ is indicative of the aerosol

absorption properties. The aerosol absorption properties and their vertical distributions play a crucial role in evaluating the direct radiative forcing of aerosols (McComiskey et al., 2008; Zarzycki and Bond, 2010). Moreover, the heating effect of the absorbing aerosol on the atmosphere results in an increase of atmospheric stability and a reduction of atmospheric vertical exchange, which further aggravates the accumulation of pollutants (absorbing particles) and a positive feedback is established. The vertical profile of LR can reflect the vertical variations of aerosol absorption properties, which can provide a

basis for studying the aerosol radiative forcing and the causes of pollution (Mishchenko et al., 2004).

LR can be obtained by a variety of methods, such as high spectrum resolution lidar (HSRL), Raman lidar and joint retrieval using sun photometer and elastic lidar (Zhao et al., 2018). Raman lidar can independently retrieve the extinction coefficient and backscatter coefficient of aerosols and obtain LR by combining elastic backscatter and Raman backscatter signals (Ansmann et al., 1992a), which is the most widely used independent measurement method at present. Moreover, the

LR measured by Raman lidar are useful indexes to study the variations of aerosol physical properties (Ferrare et al., 2001).

A large number of observations and analyses of LR have been carried out all over the world based on Raman lidar. Since the establishment of the European Aerosol Research Lidar Network (EARLINET) in 2000, a long-time series observation data of vertical distribution and LR for various types of aerosol has been obtained in the European continent (Müller et al., 2007; Wandinger et al., 2016). In South Korea and Japan, some researches have also been carried out on the

LR of the Asian dust and biomass burning aerosols based on Raman lidar (Murayama et al., 2004; Noh et al., 2007; Noh et al., 2008). The LR observed around the world usually show different values due to different types of aerosols. However, long-term observations and researches of LR in China are limited (Wang et al., 2016) due to the limitation of observation





instruments. In particular, the observations and studies of aerosol LR are still blank in East China. Generally, the range-resolved LR profiles based on independent measurement on a regional scale were very important. On the one hand, the range-resolved LR profiles are not only suggested to correct the LR of CALIPSO to improve the inversion of extinction (Papagiannopoulos et al., 2016), but also can provides a reliable basis for the inversion hypothesis of elastic lidar in Shanghai and surrounding areas, and improves the products reliability from elastic lidar network such as the Asian dust and aerosol lidar observation network. On the other hand, the vertical distribution of aerosol absorption properties reflected by LR can be used as an input parameter of regional climate models (Mehta et al., 2018), which can further improve the calculation accuracy of radiative forcing.

In addition, studying the influencing factors of LR in Shanghai can be conducive to understanding the LR variation characteristics and determining the source of pollutants. With these motivations, the vertical and temporal variations of LR and their influencing factors were analyzed in Shanghai by using the results retrieved from Raman lidar, which laid a solid foundation for the quantitative study of pollution variations and its causes in the future.

## 2 Data and methods

### 2.1 Observation equipment and data

### 2.1.1 Raman depolarization lidar

The Raman depolarization lidar (Raymetrics S.A., Athens, Greece, model LR231-D300) used in this study is deployed on a building roof (121.4333°E, 31.1916°N; 67 m above sea level, ASL) in Shanghai downtown. The site is 35 km from the East China Sea coastline, surrounded by populated residential and commercial areas. The laser of the lidar system is Nd: YAG pulse laser equipped with water cooling device. The system can emit 355 nm, 532 nm and 1064 nm laser pulses with a pulse width of 5.4 ns and a repetition rate of 20 Hz. The lidar detection system can receive 355 nm, 532 nm and 1064 nm elastic scattering signals. The 532 nm elastic scattering channel has two polarization channels, 532P (parallel) and 532S (perpendicular). In addition, the lidar can also receive the vibrational Raman scattering signals of nitrogen (387 nm) and water vapor molecules (408 nm) with an incident wavelength of 355 nm. The transient recorder can detect signals in two modes: analog and photon counting. The analog mode is suitable for detecting the strong signals at the low altitude, and the photon counting mode is suitable for detecting the weak signals at the high altitude. In order to better combine the advantages of the two modes to retrieve aerosol optical properties, it is necessary to glue the signals of the two modes (Newsom et al., 2009; Walker et al., 2014). The range resolution of the raw signal is 7.5 m, and the temporal resolutions are 1 min (before January 25, 2019) and 2 mins (after January 25, 2019), respectively. The time in the paper without special explanations was UTC.

Since the Raman Lidar used in this study can detect the Raman scattering signal of 387nm nitrogen and the Raman signals are very weak in the daytime, the LR at 355nm at night can be obtained through the retrieval. The retrieval results of



the raw signals were counted by hour, and the hours with more than 15 minutes of retrieval results were regarded as effective

observation hours. The retrieval results within the effective observation hours were averaged to obtain hourly average data. During the observation period, data of 667 effective observation hours was obtained through retrieval and statistics. The monthly distribution is shown in Fig. 1.

### 2.1.2 HYSPLIT-4

HYSPLIT-4 (Hybrid Single Particle Lagrangian Integrated Trajectory Model, Version 4) is a professional model jointly

developed by the National Oceanic and Atmospheric Administration (NOAA) and the Australian Bureau of Meteorology for the calculating and analyzing the transport and diffusion trajectories of atmospheric pollutants, and has been widely used in many studies around the world (Huang et al., 2012; Noh et al., 2007). It supports the input of a variety of meteorological data, and NOAA reanalysis of meteorological data was used in this study.

### 2.1.3 MERRA-2

MERRA-2 (The Modern-Era Retrospective Analysis for Research and Applications, Version 2) is an atmospheric reanalysis dataset provided by the National Aeronautics and Space Administration (NASA) and the Global Modeling and Assimilation Office (GMAO) (Gelaro et al., 2017). The aerosol optical property data used in this study is derived from the 1-hour average product of the MERRA-2 tavg1_2d_aer_Nx dataset, and CO column concentration from the MERRA-2 tavg1_2d_chm_Nx. The spatial resolution of the two datasets is 0.625°×0.5°. During data processing, the 24-hour data of each day was averaged

to obtain the daily average data.

### 2.1.4 ERA5

ERA5 is a global atmospheric reanalysis dataset provided by the European Centre for Medium-range Weather Forecasting (ECMWF) (Zhao et al., 2020). The RH data of ERA5 used in this paper is divided into 37 layers vertically (1 hPa–1000 hPa). The temporal resolution is 1 h and the spatial resolution is 0.125°×0.125°. In order to ensure the spatial consistency, the RH

data (121.375° E, 31.25° N) closest to the location of Raman lidar was used.

### 2.2 LR Retrieval methods

After preprocessing the original signals (D'Amico et al., 2016; Mattis et al., 2016), the lidar equations for elastic and inelastic (nitrogen Raman) backscattered signals are (Ansmann et al., 1992a; Ansmann et al., 1992b):

$$\mathrm{P}_{\lambda_0}(r) = \frac{C_1}{r^2}[\beta_{\lambda_0}^{aer}(r) + \beta_{\lambda_0}^{mol}(r)] \times \exp\{-2\int_0^r [\alpha_{\lambda_0}^{aer}(\xi) + \alpha_{\lambda_0}^{mol}(\xi)]d\xi\} \tag{2}$$

$$\mathrm{P}_{\lambda_R}(r) = \frac{C_2}{r^2} N_R(r) \frac{d\sigma_R(\pi)}{d\Omega} \times \exp\{-\int_0^r [\alpha_{\lambda_0}^{mol}(\xi) + \alpha_{\lambda_0}^{aer}(\xi) + \alpha_{\lambda_R}^{mol}(\xi) + \alpha_{\lambda_R}^{aer}(\xi)]d\xi\} \tag{3}$$



where $P_{\lambda_0}(r)$ and $P_{\lambda_R}(r)$ are the return signals from distance $r$ at the laser wavelength $\lambda_0$ and the Raman wavelength $\lambda_R$; $C_1$ and $C_2$ are the lidar system constants, including the optical transmittance and reflectivity of the dichroic plate and the filter, the overlap factor and the photoelectric conversion efficiency of the photoelectric device; $\beta_{\lambda_0}^{aer}(r)$ and $\beta_{\lambda_0}^{mol}(r)$ are the backscatter coefficients of aerosols and atmospheric molecules; $N_R(r)$ is the molecule number density of the Raman-active gas; $\dfrac{d\sigma_R(\pi)}{d\Omega}$ is the range-independent differential Raman cross section for backward direction; $\alpha_{\lambda_0}^{aer}(\xi)$ and $\alpha_{\lambda_0}^{mol}(\xi)$ are the extinction coefficients of aerosol and atmospheric molecules at $\lambda_0$; $\alpha_{\lambda_R}^{aer}(\xi)$ and $\alpha_{\lambda_R}^{mol}(\xi)$ are the extinction coefficients of aerosol and atmospheric molecules at $\lambda_R$. The 1976 American Standard Atmosphere Model is usually used to calculate the optical parameters of atmospheric molecules. The aerosol extinction coefficient can be obtained via the nitrogen Raman signals, as given by

$$\alpha_{\lambda_0}^{aer}(r) = \frac{\{\dfrac{d[\ln \dfrac{N(r)}{r^2 P_{\lambda_R}(r)}]}{dr} - \alpha_{\lambda_0}^{mol}(r) - \alpha_{\lambda_R}^{mol}(r)\}}{1 + \left(\lambda_0 / \lambda_R\right)^{-A}} \tag{4}$$

Where $A$ is the aerosol extinction Ångström index, usually with a value of 1.

The aerosol backscatter coefficient is derived from the ratio $\dfrac{P_{\lambda_0}(r)P_{\lambda_R}(r_0)}{P_{\lambda_0}(r_0)P_{\lambda_R}(r)}$, as given by

$$\beta_{\lambda_0}^{aer}(r) = \beta_{\lambda_0}^{mol}(r_0) \frac{P_{\lambda_0}(r)P_{\lambda_R}(r_0)}{P_{\lambda_0}(r_0)P_{\lambda_R}(r)} \times \frac{N_R(r)}{N_R(r_0)} \times \frac{\exp\{-\int_{r_0}^r [\alpha_{\lambda_R}^{aer}(\xi) + \alpha_{\lambda_R}^{mol}(\xi)]d\xi\}}{\exp\{-\int_{r_0}^r [\alpha_{\lambda_0}^{aer}(\xi) + \alpha_{\lambda_0}^{mol}(\xi)]d\xi\}} - \beta_{\lambda_0}^{mol}(r) \tag{5}$$

$r_0$ is the reference height where the aerosol loading is low and the backscatter coefficient of aerosol at $r_0$ is much smaller than that of atmospheric molecules.

LR can be obtained by using extinction coefficient and backscatter coefficient.

$$LR = \alpha_{\lambda_0}^{aer}(r) / \beta_{\lambda_0}^{aer}(r) \tag{6}$$

The volume depolarization ratios ($\delta$) can be measured by polarization channels (Cairo et al., 1999; Freudenthaler et al., 2017).

$$\delta(r) = K \frac{P_s(r)}{P_p(r)} \tag{7}$$





$P_s(r)$ and $P_p(r)$ are the lidar signals perpendicular and parallel to the polarization direction of the emitted laser, respectively, $K$ is the calibration factor, which is 0.074.

Aerosol optical depth (AOD) is the integral of aerosol extinction coefficient in the vertical direction (Zhao et al., 2018).

$$AOD = \int_{r_1}^{r_2} \alpha^{aer}(\xi)d\xi \tag{8}$$

$r_1$ and $r_2$ are the starting height and the ending height, respectively.

There are many sources of errors in the retrieval results. The relative errors of particle extinction coefficients caused by the assumed air density profile are 1.5% (Masonis, 2002), and the relative errors of particle backscatter coefficients caused by the reference height can be 10% (Ansmann et al., 1992a). The mean deviations of particle extinction coefficients caused by signal detection are within 15% in the 350 m–2000 m height range and within 20% in the 3000 m–4000 m height range

(Pappalardo et al., 2004). The difference is caused by the different signal-to-noise ratio at low altitude and high altitude. Due to the low signal-to-noise ratio, there are usually more missing values at the high altitudes.

## 3 Results and discussions

### 3.1 LR temporal and vertical variations

#### 3.1.1 General variation of LR

Figure. 2(a) shows the average profile of LR from 667 hours of statistical results. Because of the variability of aerosol particle size and microphysical properties with height (Singh et al., 2005), the average LR were characterized by a large variability, ranging from 17 sr to 82 sr. The LR reached the maximum at the height of 600m, and decreased with the increase of height. The average LR in the height range of 0.5 km–5 km was 41.0±22.5 sr, and relatively small of 24.8±13.7 sr above 2 km. This discrepancies were in good agreement to the results of Hee et al. (2016) in Malaysia with mostly less than 25 sr in

the altitude range of 2 km to 3 km.

In order to investigate the variations of LR and aerosol type at different altitude ranges, Fig. 2(b) presents the average LR at different altitude ranges. The average LR from 0.5 km to 1km was 68.2±19.5 sr, which was in good agreement with the 355nm LR observed by Ferrire et al. (2001) in Oklahom, America. Table 1 lists the previously obtained LR at 355nm using Raman lidar. The LR of Continental, urban and biomass burning aerosols at 355nm were generally more than 60 sr,

that of dust aerosols were 40 sr–55 sr, and that of clean marine aerosols were smaller, usually between 20 sr and 35 sr. The LR depend on the aerosol size distribution and refractive index of aerosol particles (Takamura et al., 1994; Young et al., 1993). The small LR may be caused by the shape effect of aerosol particles (not obviously nonspherical) and the relatively low absorption efficiency (Tesche et al., 2007). According to the observation in Table 1, the types of aerosols below 1km in Shanghai were mainly originated from continental, urban and biomass burning emission. The mean value of LR was

between 40 sr and 50 sr in the altitude range of 1 km–2 km, implying that the main aerosol type in that altitude range was



usually dust aerosol combined with Table 1. Above 2km, the mean values of LR were usually less than 40 sr, which alluded to an increasing influence of background aerosol (i.e. less absorbing coarse-mode particle) (Hänel et al., 2012).

Furthermore, the slope of LR for the different height ranges as shown in Fig. 2(b) also gradually decreased with the increased height. Below 3 km, LR decreased rapidly with the increase of altitude with the largest slope of -17.83 below 1.5 km. The reasons for this were that the inversion of temperature in the low layer of the planetary boundary layer (PBL) at night weakened the vertical movement of the atmosphere and inhibited the diffusion of pollutants emitted by human activities such as vehicles and fossil fuel combustion. The accumulation of pollutants in the low layer of the PBL resulted in significant differences of aerosol vertical distribution and a rapidly decreased aerosol extinction coefficient in the PBL (Liu et al., 2017; Wang et al., 2020). However, above 3km, the low aerosol concentration and homogeneous vertical distribution of aerosol led to the small differences in LR at different height ranges.

Figure. 3 shows the frequency distribution of LR in the different altitude range. Overall, LR were widely distributed in the altitude range of 0.5 km–5 km, which illustrated that aerosol types were diverse, including almost all aerosol types in Table 1. The diversity of aerosol types was due to the location of Shanghai in the main aerosol source areas (e.g. local emissions of the Yangtze River Delta region) or downwind of the source areas (e.g. long-range transport of dust and biomass burning) (Cheng et al., 2015). In most cases (about 90%) LR ranged from 10 sr to 80 sr with the highest frequency of 17.3% between 40 sr and 50 sr, indicating the dominant influence of dust on the optical properties in Shanghai. It should be noted that the number of observations trailed off at larger LR and the frequency of abnormally large LR (> 90 sr) was about 4%, which usually corresponded to biomass burning aerosol and aged forest fire aerosols (Amiridis et al., 2009; Hee et al., 2016). Due to the diverse sources of aerosols within 0.5 km–2 km, LR also had a wide distribution range, and the frequency of 40 sr–50 sr was the highest (24.6%), which was similar to the range of 0.5 km–5 km. Large LR (> 60 sr) were mainly distributed in the range of 0.5 km–2.0 km, suggesting that the aerosol in this height range had strongly absorbing ability. Although there are a few large LR (> 60 sr) above 2 km, LR was mainly distributed between 0 sr–40 sr with the highest frequency of 34% between 10 sr and 20 sr. Furthermore, The LR between 40 sr and 60 sr accounted for about 10%, which revealed that there were also some dust aerosols above 2 km.

**3.1.2 Temporal variations of LR**

Figure. 4(a) presents the seasonal variations of LR over Shanghai during the observation period. The seasonal average LR was the largest with 47.6±25.1 sr in autumn and the lowest of 39.1±19.6sr in spring. Generally, the LR of aerosol particles with stronger absorption ability is larger (Müller et al., 2007). Black carbon (BC) has a strong ability to absorb visible light (Chow et al., 2009), and aerosols rich in BC tend to show larger LR, such as biomass burning aerosols (Giannakaki et al., 2016). Shanghai area is easily affected by the smoke produced by the burning of crop residues during the harvest season of autumn (Xu et al., 2018). According to the results of Wang et al. (2014), the BC concentration in Shanghai was the lowest in spring and higher in autumn and winter. This indicated that the aerosol absorbing ability in spring was weaker compared with that in autumn and winter.



Statistics of the mean LR at different height range in each month were shown in Fig. 4(b). It can be addressed that LR

of all months decreased with the increase of the altitude. The average LR below 2 km was the largest in October, which was attributed to the smoke aerosols produced by biomass burning in the surrounding cities and rural areas during the harvest season (Nie et al., 2015). In view of vertical variations of LR in different months, the aerosols with LR > 40sr were confined within 1.5 km from March to July, while the other months were 2 km, especially in October and November within 2.5 km. This revealed a point that the diffusion heights of aerosols with absorbing properties are characterized by a strong seasonality.

In spring and summer, the diffusion height of absorbing aerosol was lower, on the contrary, it was higher in autumn and winter. The discrepancies in seasonal height distribution of aerosols can be attributed to the monsoon climate (He et al., 2006; Liu et al., 2020; Wang et al., 2016). Shanghai, on the southeast coast of China, was affected by the subtropical monsoon climate and the prevailing winds are southeast and northwest in spring and southeast in summer, respectively (Cai et al., 2010). Clean air from the sea could reduce air pollution to a large extent by diluting the concentration of pollutants (Wang et

al., 2014). In particular, the LR in March and April above 2.5 km were higher than those in other months, which may be due to the influence of dust aerosols brought by the prevailing northwest wind in spring. Liu et al. (2020) pointed out that the dust aerosols from Inner Mongolia and Gobi desert had a high frequency in spring at an altitude of 3 km–5 km in Shanghai. The prevailing winds in autumn and winter were northeast and northwest, respectively. The absorbing aerosols originated from the north China resulted in relatively large LR at high altitude in Shanghai.

Liu et al. (2012) reported that the vast majority aerosol particles in the Yangtze Delta region (including Shanghai) were below 2 km. In order to analyze the variation characteristics of LR in Shanghai more precisely, Fig. 5 shows LR of 667 effective observation hours below 2 km. The abnormally large LR (> 80 sr) were usually distributed in the PBL, meaning that local emissions were the main source of strongly absorbing aerosols. From this figure, one can also conclude that the number of larger LR within the PBL followed a decreasing trend with the passage of observation time, alluding to a

gradually reduced emission of absorbing aerosols in Shanghai. This reduction was in good agreement with the reductions of BC and particulate matter (PM) concentrations caused by a series of energy-saving and emission-reduction measures such as the Shanghai Clean Air Action Plan (2018–2022) implemented by the Shanghai government in recent years (Wei et al., 2020).

### 3.2 Analysis of influencing factors of LR

### 3.2.1 Reasons for LR variations with height

From Eq. (1), we find that LR are negatively correlated with $P(\pi)$, and the nonspherical geometry of particles can cause a reduction of the $P(\pi)$ (Müller, 2003). $\delta$ can reflect the degree of regularity of the particle shape, and are usually defined as the ratios of vertical backscatter coefficients and parallel backscatter coefficients (Behrendt and Nakamura, 2002). The smaller the value of $\delta$, the more regular the shape of the particles, i.e., the closer the particle shape is to the spherical

(Gobbi, 1998). Since the Raman depolarization lidar used in this study can only detect the polarization information of 532





nm channel, $\delta$ at 532 nm were used to analyze the sphericity of particles. It can be seen from Fig. 6(a) that LR were proportional to $\delta$ with a correlation coefficient of 0.86, which was consistent with Reagan's (1988) study that LR increased with the increase of asphericity of particles. LR is the ratio of the extinction coefficient and the backscatter coefficient. Generally, there are two reasons for the increase of LR with the increasing asphericity of particles. On the one hand, the

backscatter coefficient decreases significantly with the increase of particle asphericity, on the other hand, the extinction coefficient is sensitive to the cross-section of the particle and is less affected by the shape of the particle. In order to explore whether the decrease of LR with increasing height is affected by the particle shape effect, Fig. 6(b) shows the average profile of $\delta$ at lidar observation time. The average and median of $\delta$ gradually decreased with the increase of height, indicating that the shape of particles became more regular with the increase of height. This result evidenced the inferences of Tesche et

al. (2007) that the regular particle shape was one of the reasons for the smaller LR at high altitudes.

### 3.2.2 Influences of aerosol sources on LR

In an effort to further research the influences of wind directions on LR and its vertical distribution, cluster analysis of back trajectories was used to study the transport of atmospheric aerosols. Based on the HYSPLIT-4 model (Franke et al., 2001; Noh et al., 2007), a total of 667 backward trajectories at 1000 m altitude beginning at 72 hour before the time of Raman lidar

performance reaching the observation site were shown in Fig. 7(a). The cluster analysis resulted in 4 main air mass directions (Hänel et al., 2012; Pietruczuk and Podgorski, 2009).

The mean LR and $\delta$ between 0.5 km and 5 km of the four clusters are shown in Fig. 2(b), and Fig. 7(c) presents the distribution of LR and $\delta$ at different heights of the four clusters. The mean LR (38.7±24.2 sr) and $\delta$ (0.030±0.021) of air mass 1 were the lowest in all clusters. As air mass 1 came from Western Pacific Ocean, it could bring abundant marine

aerosols. The sea salt particles are characterized by coarse mode, which are spherical in wet conditions. In addition, marine aerosols have lower LR than dust aerosols because of their weak absorption ability (Papagiannopoulos et al., 2018). Interestingly, according to the statistics in Table 1, the LR at 355nm of clean marine aerosols were usually between 20 sr and 35 sr, which is smaller than that of aerosols brought by air mass 1 in the study. It was observed that the average LR in the range of 0.5 km–1 km was more than 60 sr, which suggested that the clean marine aerosols from the sea might be mixed

with local absorptive aerosols in Shanghai (Franke et al., 2001; Müller et al., 2007).

The average LR affected by the aerosols brought by air mass 2 was approximately equivalent to the LR affected by air mass 1, with an average value of 39.4±19 sr at 0.5 km–5 km altitude. The source region of air mass 2 was located in the Inner Mongolia, which could bring dust aerosols. During the transport over the ocean, the clean marine aerosols mixed with dust aerosols led to the LR larger than that of the clean marine aerosols.

The mean LR of air mass 3 was 44.2±24.7 sr, largest in the four clusters. As the air mass 3 passed through the northern China region where the pollution level and the pollution amount were relatively high, the aerosol particles brought by the air mass had strongly absorbing ability. In addition, it was evident that the $\delta$ corresponding to the air mass 3 below 2.5 km



were larger than that of the other three air masses in Fig. 7(c). The larger $\delta$ hinted at a high contribution of irregularly shaped aerosol particles. Air mass 3 passing through the dust source areas of Mongolia and Inner Mongolia could bring

abundant dust aerosols to the Shanghai (Huang et al., 2012). The dust aerosols usually show larger particle depolarization ratios than other types of aerosols due to the irregular particle shape (Kai et al., 2008; Murayama et al., 1999). For example, Huang et al. (2012) found that the aerosol depolarization ratios at 532 nm in the case of dust pollution were significantly greater than that in the cases of secondary inorganic pollution and biomass burning pollution. As a consequence, the larger $\delta$ corresponding to air mass 3 were attributed to dust aerosols, and the long-range transported dust aerosols could reach a

height of about 2.5 km.

The average LR affected by aerosols from air mass 4 was 42.6±21.8 sr. It should be noted that the average $\delta$ affected by aerosols brought by air mass 4 was small, which was comparable to that of air mass 1, indicating a high contribution of spherical aerosol particles, but LR was larger. By observing the trajectory of air mass 4, we found that it passed through Hubei Province with high industrial level (Wang et al., 2016) and Anhui Province with heavy pollution of biomass burning

(Wu et al., 2020). The accompanying industrial and smoke aerosols, which are approximately spherical (Giannakaki et al., 2016; Müller et al., 2007) could be responsible for the smaller $\delta$.

In summary, the variations and vertical distribution of LR and $\delta$ in Shanghai were caused by the synthetic impacts of long-range transport from different source areas and local emissions. The larger LR and $\delta$ are the results of smoke and dust aerosols from the northwest, and the mixing of aerosols accompanied by air masses from the sea and locally emitted

absorbing aerosols resulted in the smaller $\delta$ and the slightly larger LR than that of clean marine aerosols.

### 3.2.3 Influence of atmospheric turbidity on LR

AOD is an important parameter to characterize the optical properties of aerosols, which can reflect the content of aerosols in the atmosphere, and also is an important index to evaluate atmospheric quality and visibility (Cheng et al., 2015; Hess et al., 1998). Previous studies have shown a positive correlation between AOD and LR by analyzing the average LR in different

AOD ranges (Ferrare et al., 2001; He et al., 2006), due to the increase of aerosol absorption and extinction caused by the increase of small particles (Takamura et al., 1994). In addition, although some studies have analyzed the vertical profiles of LR in different pollution degree cases, the main concern was the average LR of the aerosol layer (Chen et al., 2014; Wang et al., 2016). As mentioned previously, the vertical variations of absorbing aerosols and their influencing factors played an important role in evaluating the radiation effect of aerosol and studying the cause of pollution (Mishchenko et al., 2004).

Consequently, it is meaningful to study the vertical variations of LR under different atmospheric turbidity.

The AOD was obtained by integrating the 355nm extinction coefficients in the range of 0.5 km–2 km. The average profile of LR below 2 km in different AOD ranges was drawn as shown in Fig. 8. Under clean condition, the LR decreased more dramatically with increase height. By contrast, the lack of significant vertical variability of LR in the case of high atmospheric turbidity illustrate the homogenous vertical distribution of absorbing aerosol. The result that the vertical slope of





LR presented a decreasing trend with the increasing atmospheric turbidity can be explained by aerosol radiative effects on
thermal structure and atmospheric stability. In the case of high atmospheric turbidity, aerosol particles that absorb a large
amount of solar radiation during the day radiatively warm the surface at night, but radiatively cool the air above the surface
(Jacobson and Kaufman, 2006; Ramanathan et al., 2005). The decrease in the atmosphere stability due to the temperature
difference increase vertical turbulence and results in the homogeneous vertical distribution of aerosols. On the contrary, in

the clear and pollution-free nights, the surface radiation cooling results in temperature inversions near the ground. The stable
atmosphere is not conducive to the lifting of the absorbing aerosols, resulting in a large vertical variation of LR.

**3.3 The main aerosol types causing the abnormal variation of LR**

As shown in Fig. 5, abnormally large LR occurred occasionally in relative high location approximately above the top of PBL
in spite of an usually decay trend of LR with height. To investigate the reason, we selected five days with LR > 80 sr over 1

km. According to previous reports, biomass burning aerosols are relatively small and spherical, and their strong absorption
ability makes them have large LR (Papagiannopoulos et al., 2018). For example, Amiridis et al. (2009) observed the smoke
plume from biomass burning over Greece and found that the 355nm LR ranged from 40sr to 100sr. Giannakaki et al. (2016)
used Raman lidar in South Africa and found that the 355nm LR of biomass burning aerosols was 92±10 sr. Generally, the
determination of aerosol type with large LR observed by Raman lidar was based on fire data and backward trajectory model

such as HYSPLIT (Hee et al., 2016; Noh et al., 2008). In our study, the spatial distribution of biomass burning tracers were
used to determine whether the abnormally large LR was related to biomass burning aerosols (Huang et al., 2012), which
could lay a foundation for future research on aerosol three-dimensional spatial distribution and pollution causes.

Biomass burning is one of the important sources of PM, organic carbon (OC) and black carbon (BC) in the atmosphere
(Wu et al., 2020). It also emits pollutant gases such as CO, SO2, NOx and HCN (Andreae and Merlet, 2001; Kalluri et al.,

2020; Randel et al., 2010). CO can be used as a tracer for biomass burning. For example, Huang et al. (2012) found that the
CO column concentration in the biological combustion zone was significantly different from that in the non-biomass burning
zone. In addition, it would also result in high AOD and AAOD (absorbing aerosol optical depth) in the region due to the
strong absorption of biomass burning aerosols (Shaik et al., 2019). For example, He et al. (2015) found that the AOD at
500nm increased from 0.73 to 1.00 when analyzing the smoke plume of biomass burning in Shanghai. Similarly, Vadrevu et

al. (2011) found that the AOD of wheat dregs burning season and rice dregs burning season were both high by using satellite
data in India, which were 0.598 and 0.58, respectively. CAOD is the optical depth of carbon aerosol, which is the sum of the
optical depth of black carbon and organic carbon aerosol. Hence, AOD, AAOD, CAOD at 550nm, and CO column
concentrations were used as tracers to determine whether these five cases were affected by biomass burning. Figure. 9
depicts the spatial distribution of four tracers in five cases. The AOD in Shanghai during these five days ranged from 0.45 to

1.05, pointing to the heavy aerosol pollution. The average value of AAOD was between 0.0375 and 0.1, and CAOD was
between 0.12 and 0.24, which reasonably demonstrate the presence of absorbing aerosols and carbon aerosols in Shanghai
(Shaik et al., 2019). In addition, the CO column concentrations in Shanghai for these five days were relatively high, all more





than $9\times10^{-4}$ kg/m$^2$, which illustrate the significant possibility of smoke advection. Therefore, it could be inferred that Shanghai was affected by the biomass burning aerosols from local rural areas or neighboring provinces in the five cases.

Even if the abnormally large LR above 1km was mainly relevant for the advection of biomass burning aerosols, it should be noted that the increasing aerosol extinction caused by the increase of RH could also result in the large LR (Salemink et al., 1984). For example, Ackerman (1998) found that the LR of continental aerosols increased from 40 sr to 80 sr with RH. Figure. 10 presents the LR and RH profiles at three times. Above 1km, LR was a function of RH, and the abnormally large LR had a good corresponding relationship with the high RH, which demonstrated that the abnormally

larger LR above 1km was also related to the high RH.

## 4 Conclusions

For the first time, long-term (2017–2019) observation of Raman lidar was carried out in Shanghai. The aerosol LR at 355nm were retrieved, and the variations of LR and their influencing factors were analyzed in detail based on 667-hours data. In the height range of 0.5 km–5 km, about 90% of LR were distributed in 10 sr–80 sr, with an average of 41.0±22.5 sr, and LR

decreased with the increase of height. The average LR in autumn was the largest, which was 47.6±25.1 sr. The LR in summer and winter were close, 41.0±21.6 sr and 42.0±27.3 sr, respectively, and the LR in spring was the smallest. The seasonal variations of LR was closely related to the seasonal variations of BC concentration. In the height range of 0.5 km–2 km, the average LR was the largest in October, which was relevant for the biomass burning aerosols produced by burning straw in the surrounding cities and rural areas during the harvest season. In addition, affected by the prevailing winds in

spring and summer, the aerosols with LR > 40sr were confined within 1.5 km from March to July.

      LR and $\delta$ were positively correlated, meaning that the more regular the particle shape, the smaller the LR. $\delta$ decreased with the increase of height, which proved that the particle shape was one of the factors affecting the vertical distribution of LR. LR had a strong dependence on the source directions of the air masses. Large LR coincided with the air masses from the northwest, while the air masses from the east led to small LR. In addition, the shape of aerosol particles was

the most irregular due to the aerosols brought by air masses from the northwest. The vertical distribution of LR was affected by the atmospheric turbidity, the smaller the AOD, the greater the vertical change of LR.

      For the abnormal change which was different from an decay trend of LR with height, we analyzed the spatial distribution of 500 nm AOD, AAOD, CAOD and CO column concentrations of five cases with LR >80 sr over 1km, and found that Shanghai was located in or affected by high value centers. Therefore, it could be inferred that the large LR above

1km in Shanghai were relevant for biomass burning aerosols. In addition, the large LR above 1km at some times was also related to the high RH.



**Data availability**

The data presented in this paper are available from the corresponding authors upon request.

**Competing interests**

The authors declare that they have no conflict of interest.

**Author contributions**

TL retrieved the data and wrote the paper. QH and YC formulated the project goals and edited and reviewed the manuscript. JL, QL and WG downloaded and analyzed the reanalysis data. GH and WS revised the manuscript.

**Acknowledgments**

We are grateful to the NASA for providing MERRA-2 data and the NOAA Air Resources Laboratory (ARL) for the provision of the HYSPLIT transport and dispersion model. And, we also gratefully acknowledge the ECMWF for the provision of the ERA5 dataset.

**Financial support**

This work was supported by the National Key R&D Program of China (Grant No. 2016YFC0201900); the National Natural
Science Foundation of China (Grant No. 41975029); the Science Research Project of Shanghai Meteorological Service (Grant No. MS202016); the Chinese Ministry of Science and Technology (Grant No. 2018YFC1506305); the National Natural Science Foundation of China (Grant No. 91644211), and the Fundamental Research Funds for the Central Universities (Grant No. 2232019D3-27).

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

the transport of dust aerosols in the Tianjin area. Atmospheric Environment 237.



**Table 1 355 nm LR observed by Raman lidar in different regions.**

| Region | Aerosol type | LR(sr) | Reference |
|---|---|---|---|
| Oklahom, America | Continental | 68±12 | (Ferrare et al., 2001) |
| Tokyo, Japan | Dust | 49 | (Murayama et al., 2004) |
| Anmyeon Island, Korea | Dust | 46.8±6.5 | (Noh et al., 2007) |
| | Anthropogenic | 71.1±8.2 | |
| Thessaloniki, Greece | Biomass burning | 40–100 | (Amiridis et al., 2009) |
| Granada, Spain | Biomass burning | 60–65 | (Alados-Arboledas et al., 2011) |
| Senegal | Dust | 53 | (Veselovskii et al., 2016) |
| South Africa | Urban/industrial | 41–59 | (Giannakaki et al., 2016) |
| | Biomass burning | 81–119 | |
| | Mixed | 59–90 | |
| Penang, Malaysia | Marine | 20±6–30±9 | (Hee et al., 2016) |
| | Urban | 30±9–60±18 | |
| | Wood burning | 60±18–80±24 | |
| | Aged forest fire | 80±24–120±36 | |
| south of Australia | Marine | 19±7 | (Alexander and Protat, 2019) |





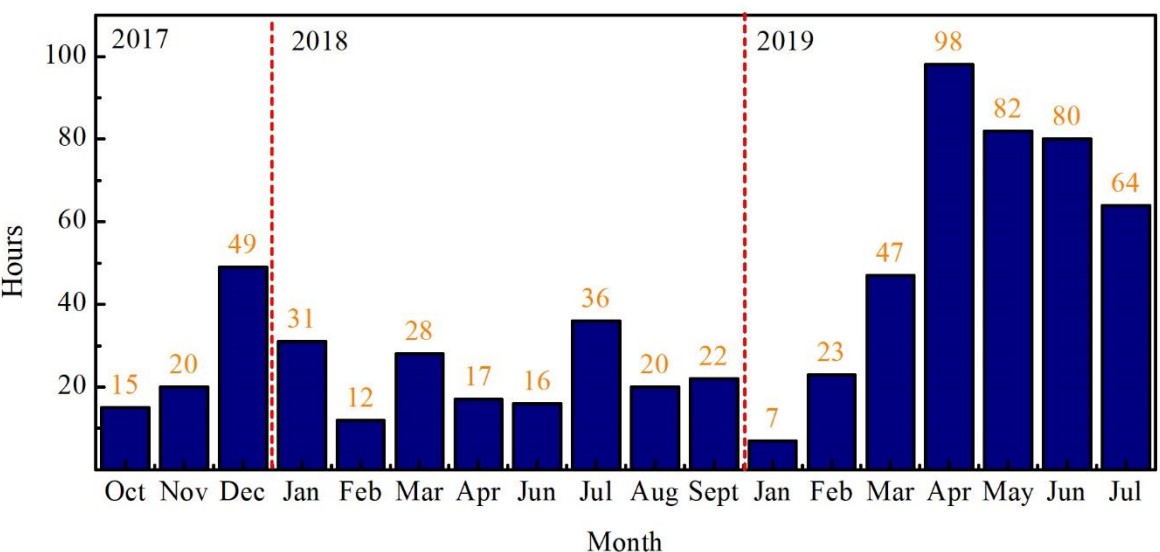

**Figure 1: Effective observation hours per month from 2017 to 2019.**

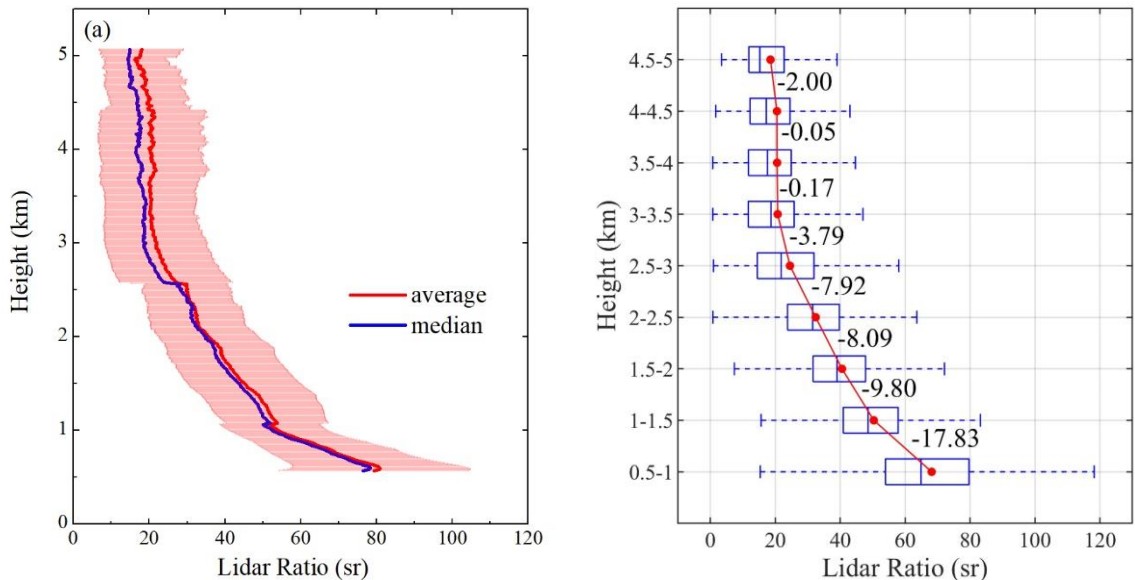


**Figure 2: General variation of LR. (a) Average profile of LR. The red line is the mean profile, the blue line is the median profile, and the red shadow is the error bar, indicating the standard deviation; (b) Average LR in different altitude range. The red line is a line of average values at different heights. The number between two points are the slopes between the two points.**





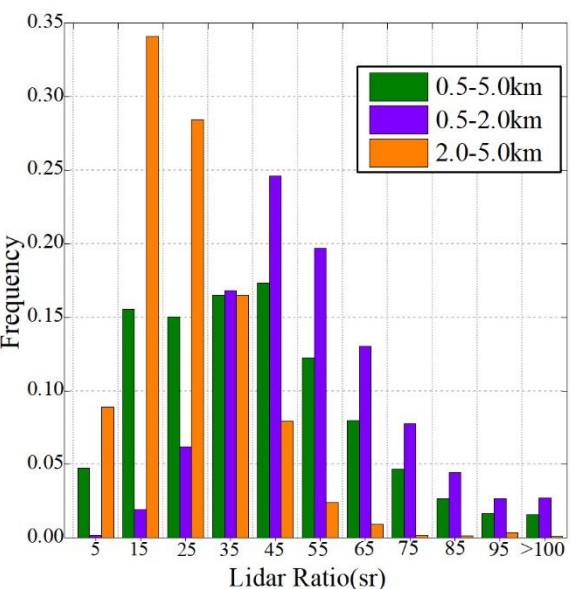

**Figure 3: LR frequency distribution.**

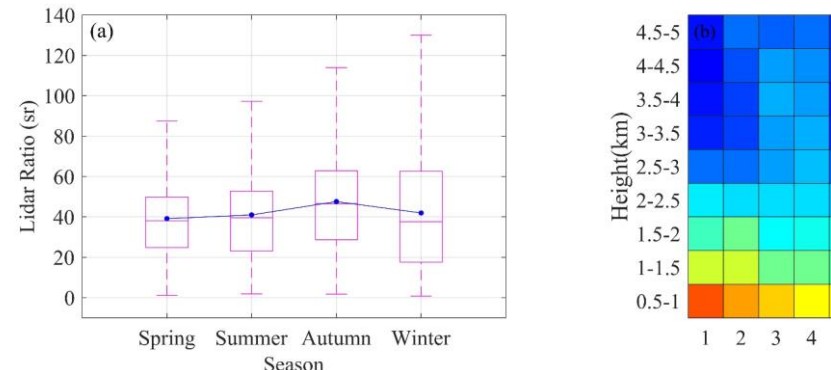

**Figure 4: LR temporal variations. (a) Seasonal variations of LR. The blue dots are the average for each season; (b) Average of LR at different altitudes in different months. The black areas indicate invalid value.**


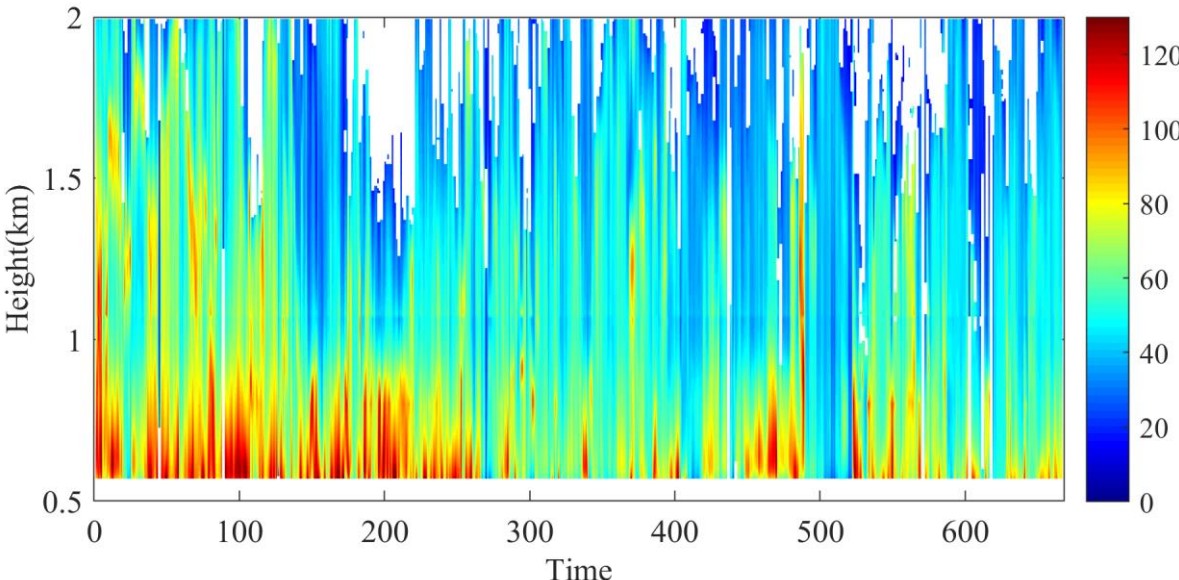

**Figure 5: The average LR of effective observation hours. The white areas indicate invalid values.**

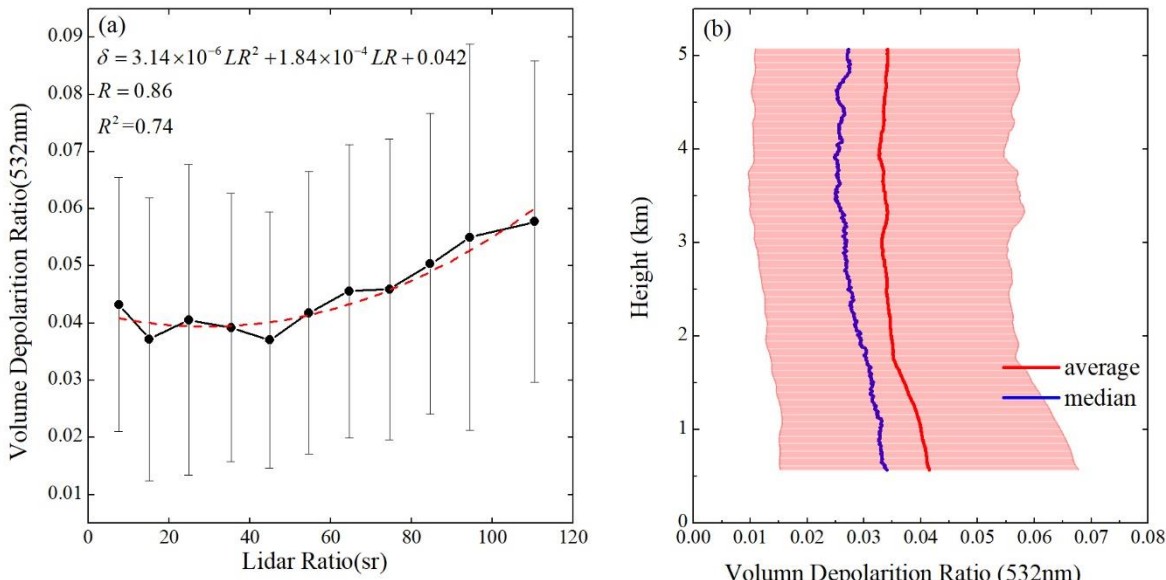

**Figure 6: Effect of $\delta$ on LR. (a) LR was divided into 11 intervals and the mean value of $\delta$ in each interval was calculated. The dot and error bar represent the mean and standard deviation of $\delta$ in each interval. The red dotted line is the fitting line of LR and $\delta$; (b) The red line is the mean profile of $\delta$, the blue line is the median profile of $\delta$, and the red shadow is the error bar,**
**meaning the standard deviation.**



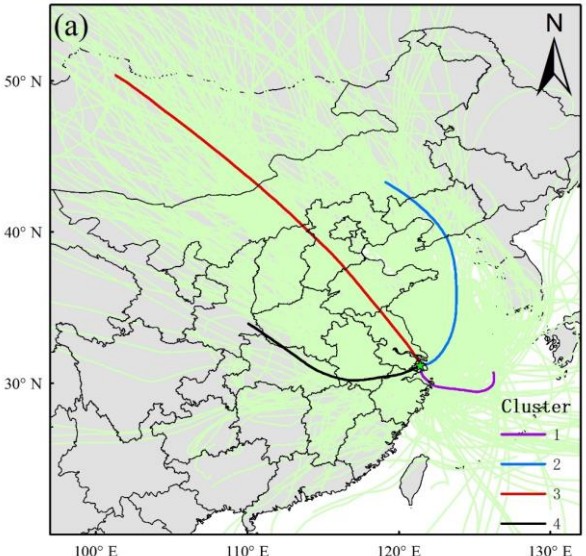

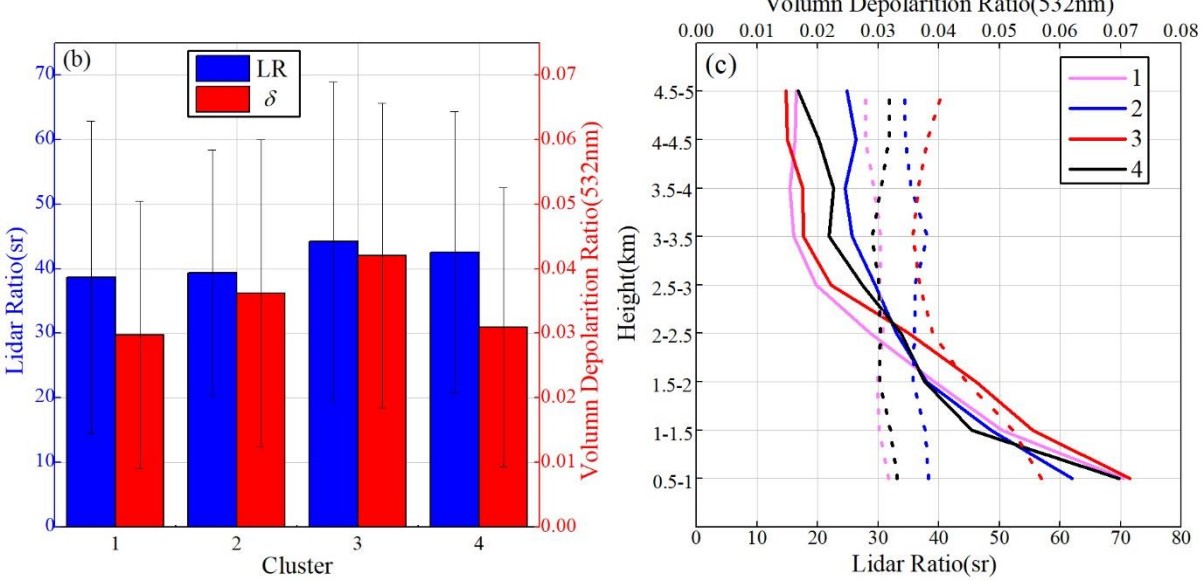

**Figure 7: (a) 72-h back trajectory cluster analysis; (b) Mean values of LR and $\delta$ corresponding to different air masses; (c) The average of LR and $\delta$ at different heights corresponding to different air masses, the solid line is LR, the dotted line is $\delta$ .**





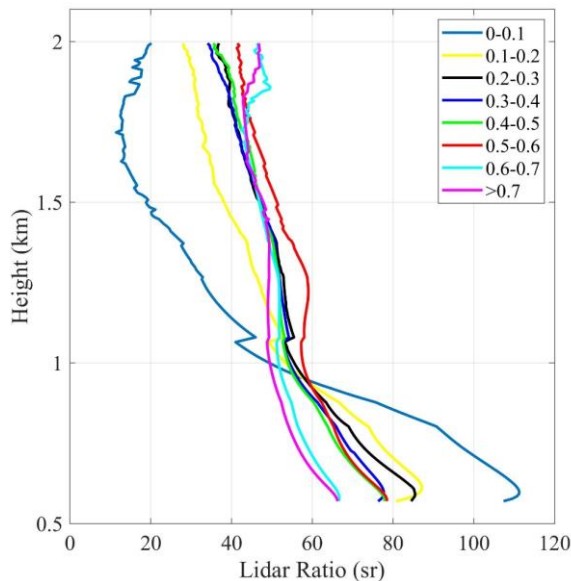


**Figure 8: LR profiles in different AOD intervals.**

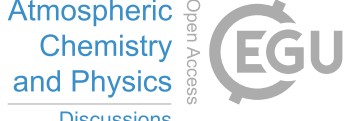

**Figure 9: Spatial distribution of AOD, AAOD, CAOD and CO column concentrations in 5 cases. From left to right, different cases are represented, and from top to bottom, different tracers are represented.**





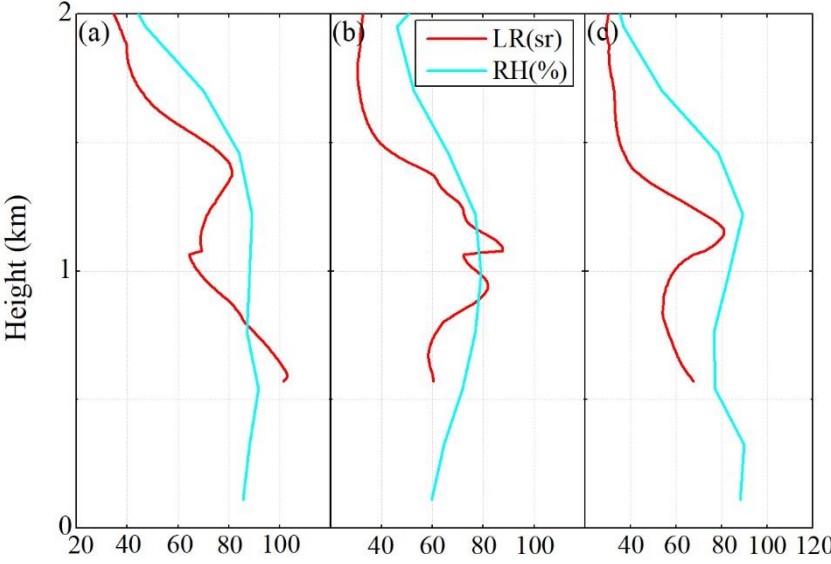


**Figure 10: RH and LR profiles at the same time. (a) December 8, 2017, 20:00; (b) March 17, 2018, 16:00; (c) September 23, 2018, 20:00.**