# Peer review of "Study on variations in lidar ratios for Shanghai based on Raman lidar"

_Atmospheric Chemistry and Physics, 2020_

## Referee Comment (RC1) · Anonymous Referee #1 · 23 Dec 2020

The aerosol types are very important information in assessing the impacts of aerosols on climate forcing. Accurate measurements of Lidar ratios (LR) and better understanding of their variation characteristics provide a way from remote sensing for the scientific research on aerosol type information. This study addresses the results of a long-term observation of LR at 355 nm of Raman lidar in Shanghai from 2017 to 2019, and analyzed their variations and influencing factors. This kind of observations about LR are rare and worth encouraging, especially over Eastern Asia region. I recommend a minor revision before publication with ACP. The detail comments or suggestions are shown below.

My general concern about this study is that although the data results are rare and good, Some of the analysis on the reasons for LR vertical and temporal variations and

aerosol source influences are not straightforward. Several inferences and methods in part 3 seems too complicated and sometimes confused. First, it is difficult to conclude aerosol type according to the comparison of LR data with some literature reports in table 1 of the manuscript. Second, the results from the usage of cluster analysis of back trajectories in 3.2.2 are not significant without a statistical test. In fact, the back-trajectory analysis should be effective for long-range transport dominated aerosols. Over an urban area of Shanghai, I suppose the status of aerosols from long-time averaged perhaps are mainly dominated by local urban emissions. Dust fraction in the aerosols should be a main factor to affect the LR data and volume depolarization ratios ($\delta$), sometimes absorbing aerosols from primary aerosols may also cause an increase of LR in the surface layer. I suggest a study to check the dust fraction on LR and $\delta$ only with the surface PM2.5/PM10 data. The PM2.5/PM10 data should be easy to obtain over urban area of Shanghai. If the authors have more chemical composition observations, for examples, EC, that will be better.

And, I have some minor and technical comments for the authors to address:

1, the English of the paper should be improved, for examples, some definite articles 'the' be misused.

2, Line 43, "The P($\pi$) was related to sphericity of the particle which can be obtained from the polarization lidar", can it be obtained from the polarization lidar?

3, Line 46, "Moreover, the heating effect of the absorbing aerosol on the atmosphere results in an increase of atmospheric stability and a reduction of atmospheric vertical exchange, which further aggravates the accumulation of pollutants (absorbing particles) and a positive feedback is established". Whether this conclusion is right, I think it is depending on a suitable vertical distribution of the absorbing aerosol. If the absorbing aerosols are on near surface layer their heating effect will enhance the instability of the atmospheric boundary layer.

4, Line 92, "the Raman signals are very weak in the daytime", are the Raman signals

stronger in the night time than in the daytime, I think they are same, but signal-to-noise ratios are different at daytime and nighttime.

5, Line 111-115, and 338-340, RH data from a model simulation or reanalysis data are not credible. I do not think this analysis is useful to the study.

6, In 3.1.1 General variation of LR. Some comparisons with those values in table 1, then conclude the aerosol type, I do not agree with this method. Dust fraction, or fine-mode fraction, maybe is more direct to help the analysis of the aerosol type.

7, Line 172, "Above 2km, the mean values of LR were usually less than 40 sr, which alluded to an increasing influence of background aerosol (i.e. less absorbing coarse-mode particle)", I think "coarse-mode particle" is not accurate. The coarse-mode particle will lead an increase of LR. Fine-mode secondary aerosols are dominated in long-range transport process over an urban area.

8, Results and analysis related to Figure 4, I do not think the current observation period of the data support a seasonal or monthly/annual change analysis. The observation period presented by Fig.1 only include some individual months over the 3 year respectively, it is not reasonable to combine the individual months from different years to an annual or seasonal change. The authors need longer and more continuous data.

9, Figure 5, I suggest date labels for the x-axis instead of sequence numbers.

10, Analysis related to Figure 7 with back trajectory cluster analysis, I do not think it is significant.

11, Analysis related to Figure 8 & 9, why not to check by some surface observations instead of only using AOD, for examples, PM2.5 mass concentration?

---

## Referee Comment (RC2) · Anonymous Referee #2 · 30 Dec 2020

Aerosol lidar ratio (LR) is a key parameter for retrieving aerosols optical properties from elastic lidar measurements, and better evaluating its climatic effects. The article presents an investigation of lidar ratio (355 nm) variation of atmospheric aerosols in Shanghai from long-term Raman/polarization lidar measurements. Moreover, relation between LR at 355 nm and other factors are discussed in detail. The topic is of sufficient interest to the communities of study of laser remote sensing and atmospheric aerosol. In general, I find this manuscript to be of interest for publication and appropriate for Atmospheric Chemistry and Physics. There are several suggestions for improvement listed below that should be considered by the authors and the editors before publication.

1. The title of the manuscript is inappropriate. In my opinion, it is preferred to use "Long-

term variation of aerosols lidar ratio in Shanghai based on Raman lidar measurement".

2. This study discuss distribution of lidar ratio at 355 nm, it is suggested that the authors have to clearly specify it throughout the manuscript, especially mark "lidar ratio (355 nm)" in all figures.

3. Honestly speaking, this study cannot provide enough valuable LR information for improvement of the CALIPSO lidar retrieval. Because CALIPSO algorithm need LR at 532nm for seven aerosols types, not that at 355nm. However, LR at 355nm in this study would be very useful for the EARTHcare lidar in the future. Please rewrite sentences in Line 64-66.

4. Line 60-61: combine citations to "Noh et al., 2007 and 2008". Similar to citations in line 118. Please check such problem throughout the manuscript.

5. Section 2: retrieved method of aerosol optical properties from Raman lidar is widely used and almost common knowledge among lidar community. The authors do not modify or improve the method in this study at all. So, it is suggested that section 2.2 could be compressed. More important information (such as lidar data correction) should be briefly introduce in section 2. For example, overlap correction is very important before retrieving LR from Raman lidar observation. Improper overlap correction would lead to large uncertainty.

6. LR is a really complicated parameter which not only depends on aerosol types. It is hard to identify aerosol type from LR only, without additional independent information. The authors claimed that dust aerosol is usually distributed around 1-2 km, according to range of LR variation. This conclusion is inconsistent with statement in line 193-194. It should be noted that depolarization ratio can identify dust from other aerosol reasonably, rather than LR. Please rewrite the sentences.

7. Page 9 line 247: I guess "an effort" should be "order".

8. Page 12 line 343: change "667-hours" to '667-hour'.

9. Figure 5: x-axis of this figure should be marked by date (not hour), so that readers easily understand seasonal variation of LR in Shanghai.

10. Figure 6: it is well known that dust aerosols usually show large depolarization ratio (DR). As descried by the authors, LR of dust is 40-60 sr. However, LR which corresponds to large DR are in range of 100-120 sr. Please explain the reason.

11. Figure 9: Please mark the location of lidar site in all panels.

12. The English of manuscript should be further improved before publication.

---

## Author Comment (AC1) · 16 Feb 2021

Anonymous Referee #1 Interactive comment on "Study on variations in lidar ratios for Shanghai based on Raman lidar" by Tongqiang Liu et al.

The aerosol types are very important information in assessing the impacts of aerosols on climate forcing. Accurate measurements of Lidar ratios (LR) and better understanding of their variation characteristics provide a way from remote sensing for the scientific research on aerosol type information. This study addresses the results of a long-term observation of LR at 355 nm of Raman lidar in Shanghai from 2017 to 2019, and analyzed their variations and influencing factors. This kind of observations about LR are rare and worth encouraging, especially over Eastern Asia region. I recommend a minor

revision before publication with ACP. The detail comments or suggestions are shown below.

Reply: We are very grateful for the referee's encouraging comments and recommendations for publication with minor revision. The following are our point-by-point responses to the referee's comments.

My general concern about this study is that although the data results are rare and good, Some of the analysis on the reasons for LR vertical and temporal variations and aerosol source influences are not straightforward. Several inferences and methods in part 3 seems too complicated and sometimes confused. First, it is difficult to conclude aerosol type according to the comparison of LR data with some literature reports in table 1 of the manuscript. Second, the results from the usage of cluster analysis of back trajectories in 3.2.2 are not significant without a statistical test. In fact, the back trajectory analysis should be effective for long-range transport dominated aerosols. Over an urban area of Shanghai, I suppose the status of aerosols from long-time averaged perhaps are mainly dominated by local urban emissions. Dust fraction in the aerosols should be a main factor to affect the LR data and volume depolarization ratios ($\delta$), sometimes absorbing aerosols from primary aerosols may also cause an increase of LR in the surface layer. I suggest a study to check the dust fraction on LR and $\delta$ only with the surface PM2.5/PM10 data. The PM2.5/PM10 data should be easy to obtain over urban area of Shanghai. If the authors have more chemical composition observations, for examples, EC, that will be better.

Reply: We thank the referee for providing insightful comments and constructive suggestions, which have been very helpful for us to improve the manuscript. The identification of aerosol type can not only use LR, but also needs other physical parameters such as depolarization ratio and color ratio. Therefore, the analysis on aerosol type identified by LR in section 3.1.1 is not accurate and we've deleted them. We conducted significance tests on cluster analysis results in section 3.2.2 according to referee's suggestions, and the difference of LR ($\delta$) between the four clusters was significant. In

addition, the backward trajectory analysis in Section 3.2.2 was mainly to explore the impact of aerosol sources on LR, so we set the airflow reaching altitude at 1000 m above sea level. Generally, aerosols above PBL are mainly affected by long-range transport. Fig. 1 shows the mean value of PM mass concentration of four clusters. The PM mass concentration data obtained from observations near the Raman lidar site. Lower PM2.5/PM10 indicates significant contributions by coarse model aerosols such as dust, and higher PM2.5/PM10 is attributed to anthropogenic aerosols such as sulfate and nitrate (Tian et al., 2018). The mean PM2.5/PM10 affected by the aerosols brought by air mass 4 is 0.63, lowest in the four clusters, which indicated that coarse mode particles contribution is significant. This is different from the result in 3.2.2 that the contribution of spherical smoke particles brought by air mass 4 is larger. The discrepancy may be due to the fact that PM data come from ground sampling, which is difficult to evaluate the impact of long-range transport aerosols above PBL on LR.

1. the English of the paper should be improved, for examples, some definite articles 'the' be misused.

Reply: We appreciated the referee's suggestions. We reviewed the manuscript carefully and have tried our best to corrected grammatical errors and improve the English of the paper.

2. Line 43, "The P($\pi$) was related to sphericity of the particle which can be obtained from the polarization lidar", can it be obtained from the polarization lidar?

Reply: We appreciated the referee's suggestions. The phase function at 180° (P($\pi$)) can't be obtained from polarization lidar, and we have revised it. P($\pi$) was related to sphericity of particles, and the sphericity information of particles can be obtained from polarization lidar.

3. Line 46, "Moreover, the heating effect of the absorbing aerosol on the atmosphere results in an increase of atmospheric stability and a reduction of atmospheric vertical exchange, which further aggravates the accumulation of pollutants (absorbing particles) and a positive feedback is established". Whether this conclusion is right, I think it is depending on a suitable vertical distribution of the absorbing aerosol. If the absorbing aerosols are on near surface layer their heating effect will enhance the instability of the atmospheric boundary layer.

Reply: We agree with referee's concerns and have revised it.

Moreover, absorbing aerosols increases atmospheric stability by reducing the solar radiation reaching the surface during the day. On the contrary, absorbing aerosols on near surface layer heats the surface and increases the atmospheric instability during night (Jacobson, 1998; Jacobson and Kaufman, 2006).

4. Line 92, "the Raman signals are very weak in the daytime", are the Raman signals stronger in the night time than in the daytime, I think they are same, but signal-to-noise ratios are different at daytime and nighttime.

Reply: We agree with referee's concerns. The intensity of Raman signals is the same in daytime and night time, but signal-to-noise ratios of Raman signals in daytime are much lower than that in night time. We've revised this sentence.

Since Raman Lidar used in this study can detect the Raman scattering signal of 387nm nitrogen and signal-to-noise ratios of Raman signals in daytime are much lower than that in night time, the 355nm LR at night can be obtained through the retrieval.

5. Line 111-115, and 338-340, RH data from a model simulation or reanalysis data are not credible. I do not think this analysis is useful to the study.

Reply: We appreciate the referee's suggestions and add some discussion about the reasons of using RH data from ERA5.

In recent years, some studies have evaluated the accuracy of reanalysis data provided by the European Center for Medium-Range Weather Forecasts (ECMWF) based on radiosonde data. For example, Luo et al. (2020) found that the average RH discrepancy between ERA-Interim radiosonde was within 10% below 500 hPa. Song et al. (2020)

found that the root mean square error (RMSE) of ERA5 RH was 3.85% compared with the RH profile of the radiosonde. The above results show that RH from reanalysis data has good accuracy, and has been widely used in various research fields (Sajadi et al., 2020; Tzanis et al., 2019; Xiao et al., 2020).

In addition, we compared the water vapor mixing ratio of Raman lidar with that of ERA5. As shown in Fig. 2, the water vapor obtained from Raman lidar has good consistency with that of ERA5, and the correlation coefficient is 0.94. The comparison between Raman lidar retrieved results and reanalysis data will be put into our next paper.

6. In 3.1.1 General variation of LR. Some comparisons with those values in table 1, then conclude the aerosol type, I do not agree with this method. Dust fraction, or fine-mode fraction, maybe is more direct to help the analysis of the aerosol type.

Reply: The referee's comments are very valuable. Dust fraction or fine-mode fraction combined with LR really helps to analyze aerosol types more reasonably. The fine-mode fraction can be determined indirectly by the ratio of extinction coefficients of two wavelengths (color ratio) (Liu et al., 2017). Our Raman lidar can only obtain accurate extinction coefficient profile at 355 nm, but cannot obtain accurate extinction coefficient profile at 532 nm due to lack of the corresponding Raman channel. Therefore, the accurate color ratio cannot be obtained, which leads to the inaccurate determination of aerosol particle size. According to definition of LR in Müller's (Müller, 2003) study, LR depends on the absorption properties and phase function of the aerosol. In fact, aerosol absorption properties and phase function can't be obtained only from Raman lidar observation at present. Therefore, we just tried to analyze possible aerosol types by comparing our LR values with those of typical aerosols in previous studies. We agree with your suggestions and have deleted the corresponding analysis on aerosol type identified by LR in section 3.1.1. In order to more accurately analyze the impact of changes in aerosol types on LR, we will use collaborative observation of drones to obtain vertical variation of aerosol absorption properties and size distribution in the future.

7. Line 172, "Above 2km, the mean values of LR were usually less than 40 sr, which alluded to an increasing influence of background aerosol (i.e. less absorbing coarse-mode particle)", I think "coarse-mode particle" is not accurate. The coarse-mode particle will lead an increase of LR. Fine-mode secondary aerosols are dominated in long-range transport process over an urban area.

Reply: We appreciate the referee's suggestions. Ferrare et al. (2001) pointed out that LR at 355 nm increased with the increase of the relative amount of fine model aerosols. The small LR at high altitude were usually related to low aerosol concentration and low absorption efficiency of aerosols (Hänel et al., 2012; Hee et al., 2016). Therefore, we've revised the sentence.

Above 2km, mean values of LR were usually less than 40 sr, which was related to low aerosol concentration and low absorption efficiency of aerosols (Hänel et al., 2012; Hee et al., 2016).

8. Results and analysis related to Figure 4, I do not think the current observation period of the data support a seasonal or monthly/annual change analysis. The observation period presented by Fig.1 only include some individual months over the 3 year respectively, it is not reasonable to combine the individual months from different years to an annual or seasonal change. The authors need longer and more continuous data.

Reply: We agree with referee's comments that it is more reasonable to use longer and more continuous data to analyze the seasonal or monthly variation of LR. However, it can be seen from Table R1 and R2 that although Raman lidar observation time is only three years, it covers all months, especially in 2018, which has been continuously observed for almost the whole year. In addition, there are enough samples in different months/seasons, and the average LR have higher statistical power. Therefore, we believe that monthly/seasonal variations of LR are still credible, and the LR monthly/seasonal variations also provides reliable observation basis for the correction of parameters from elastic lidar. Annual variation of LR wasn't analyzed due to the

short observation time. Table 1 Number of effective observation hours in different season during Raman lidar observation period. Season Spring Summer Autumn Winter Number 272 216 57 122 Table 2 Number of effective observation hours in different month during Raman lidar observation period. Month Jan Feb Mar Apr May Jun Jul Aug Sept Oct Vov Dec Number 38 35 75 115 82 96 100 20 22 15 20 49

9. Figure 5, I suggest date labels for the x-axis instead of sequence numbers.

Reply: The x-axis of Figure 5 has been marked by date (Day/Month/Year) according to referee's suggestions.

10. Analysis related to Figure 7 with back trajectory cluster analysis, I do not think it is significant.

Reply: We agree with referee's concerns and have added significance tests of the results from cluster analysis in the revised manuscript.

Backward trajectory cluster analysis based on HYSPLIT model is widely used in atmospheric aerosol research (Wang et al., 2020; Xu et al., 2018; Zhang et al., 2020). We performed a significance test on the results of the cluster analysis, and the one-way ANOVA showed that P<0.05, indicating that the LR of the four clusters were significantly different. Similarly, there were significant differences in $\delta$ among the four clusters.

In addition, we have added the percentage of each cluster in Fig 7 (a).

11. Analysis related to Figure 8 & 9, why not to check by some surface observations instead of only using AOD, for examples, PM2.5 mass concentration?

Reply: The referee's comments are very valuable. Fig. 3 shows the vertical variation of LR at different PM2.5 mass concentrations. The slope of LR vertical variation didn't show a decreasing trend with the increase of PM2.5 mass concentration. In addition, Fig. 4 shows the hourly variation of PM2.5 in five cases with abnormal LR variation. Except for case 2, the PM2.5 mass concentrations in the other four cases

were larger. PM2.5 data can well reflect surface pollution degree of particulate matter, but it is difficult to identify aerosol types, especially above 1km altitude. Therefore, spatial distribution of tracer were used to determine whether these five cases were affected by biomass burning aerosols.

[Figure]

**Fig. 1.** PM of four clusters.

[Figure]

**Fig. 2.** Comparison of water vapor mixing ratio between Raman lidar and ERA5.

[Figure]

**Fig. 3.** LR profiles in different PM2.5 mass concentration intervals.

**Fig. 4.** The hourly variation of PM2.5 mass concentration in five cases, and red dotted line is daily average value.

---

## Author Comment (AC2) · 16 Feb 2021

Anonymous Referee #2 Interactive comment on "Study on variations in lidar ratios for Shanghai based on Raman lidar" by Tongqiang Liu et al.

Aerosol lidar ratio (LR) is a key parameter for retrieving aerosols optical properties from elastic lidar measurements, and better evaluating its climatic effects. The article presents an investigation of lidar ratio (355 nm) variation of atmospheric aerosols in Shanghai from long-term Raman/polarization lidar measurements. Moreover, relation between LR at 355 nm and other factors are discussed in detail. The topic is of sufficient interest to the communities of study of laser remote sensing and atmospheric aerosol. In general, I find this manuscript to be of interest for publication and

appropriate for Atmospheric Chemistry and Physics. There are several suggestions for improvement listed below that should be considered by the authors and the editors before publication.

Reply: We are very grateful for referee's encouraging comments and recommendation for publication. We have tried our best to improve this paper according to referee's very helpful suggestions. The following are our point-by-point responses to referee's comments.

1. The title of the manuscript is inappropriate. In my opinion, it is preferred to use "Long term variation of aerosols lidar ratio in Shanghai based on Raman lidar measurement".

Reply: The title of the manuscript has been revised according to referee's suggestions.

2. This study discuss distribution of lidar ratio at 355 nm, it is suggested that the authors have to clearly specify it throughout the manuscript, especially mark "lidar ratio (355 nm)" in all figures.

Reply: We appreciate the referee's suggestions, and have specified in section 2.1.1 that the LR in our study was at 355 nm.

Therefore, the LR obtained and discussed in this study is at 355nm.

We've also marked "355 nm Lidar Ratio" in Fig. 2, 3, 4(a), 6(a), 7(b), 7(c) and 8 according to the referee's suggestions.

3. Honestly speaking, this study cannot provide enough valuable LR information for improvement of the CALIPSO lidar retrieval. Because CALIPSO algorithm need LR at 532nm for seven aerosols types, not that at 355nm. However, LR at 355nm in this study would be very useful for the EARTHcare lidar in the future. Please rewrite sentences in Line 64-66.

Reply: We agree with referee's concerns and have rewritten it.

On the one hand, the range-resolved LR obtained from the ground-based Raman lidar

can not only be used to compare with 355nm LR obtained from ATLID (Atmospheric LI-Dar) on the EarthCARE (Earth Clouds and Radiation Explorer) planned to be launched by ESA (European Space Agency) (Liu et al., 2020; Nicolae et al., 2018), but also can provides a reliable basis for the inversion hypothesis of elastic lidar in Shanghai and surrounding areas, and improves the products reliability from elastic lidar network such as the Asian dust and aerosol lidar observation network.

4. Line 60-61: combine citations to "Noh et al., 2007 and 2008". Similar to citations in line 118. Please check such problem throughout the manuscript.

Reply: We appreciate referee's reminding and have checked and revised the inappropriate citations throughout the manuscript.

In South Korea and Japan, some researches have also been carried out on the LR of the Asian dust and biomass burning aerosols based on Raman lidar (Murayama et al., 2004; Noh et al., 2007 and 2008).

Because Section 2.2 was compressed, the citations in line 118 were deleted.

5. Section 2: retrieved method of aerosol optical properties from Raman lidar is widely used and almost common knowledge among lidar community. The authors do not modify or improve the method in this study at all. So, it is suggested that section 2.2 could be compressed. More important information (such as lidar data correction) should be briefly introduce in section 2. For example, overlap correction is very important before retrieving LR from Raman lidar observation. Improper overlap correction would lead to large uncertainty.

Reply: The referee's comments are very valuable. We've compressed section 2.2 and added a brief introduction of lidar data pre-processing. Because signals in the incomplete overlap area were not used for retrieval, the overlap correction was not introduced in section 2.2.

Original signals need to be pre-processed before retrieval, including background sub-

traction, photon counting signal dead-time correction, gluing, and overlap correction (D'Amico et al., 2016). The calculation of the glue coefficients used the method proposed by Newsom et al. (2009). In order to reduce the influence of incompletely overlapping detection areas of lidar on retrieved results, only signals in the complete overlap area were used for retrieval. In addition, affected by the location altitude of Raman lidar and the least square method used in the retrieval process, the lowest height of LR obtained by Raman method is 569.5m (ASL). Since Raman Lidar used in this study can detect the Raman scattering signal of 387nm nitrogen and signal-to-noise ratios of Raman signals in daytime are much lower than that in night time, the 355nm LR at night can be obtained through retrieval. The retrieval results of raw signals were counted by hour, and the hours with more than 15 minutes of retrieval results were regarded as effective observation hours. The retrieval results within the effective observation hours were averaged to obtain hourly average data. During observation period, data of 667 effective observation hours was obtained through retrieval and statistics. The monthly distribution is shown in Fig. 1.

6. LR is a really complicated parameter which not only depends on aerosol types. It is hard to identify aerosol type from LR only, without additional independent information. The authors claimed that dust aerosol is usually distributed around 1-2 km, according to range of LR variation. This conclusion is inconsistent with statement in line 193-194. It should be noted that depolarization ratio can identify dust from other aerosol reasonably, rather than LR. Please rewrite the sentences.

Reply: We agree with referee's concerns. It is not accurate to identify aerosol types only by LR and we've deleted Table 1 and the analysis on aerosol type determined by LR in section 3.1.1.

The average LR from 0.5 km to 1km was 68.2±19.5 sr, which was in good agreement with the 355nm LR observed by Ferrare et al. (2001) in Oklahom, America. The mean value of LR was between 40 sr and 50 sr in the altitude range of 1 km–2 km, and the mean values of LR were usually less than 40 sr above 2km, which was related to low

aerosol concentration and low absorption efficiency of aerosols (Hänel et al., 2012; Hee et al., 2016).

Figure. 3 shows the frequency distribution of LR in the different altitude range. Overall, LR were widely distributed in the altitude range of 0.5 km–5 km. In most cases (about 90%) LR ranged from 10 sr to 80 sr with the highest frequency of 17.3% between 40 sr and 50 sr. It should be noted that the number of observations trailed off at larger LR and the frequency of abnormally large LR (> 90 sr) was about 4%. LR also had a wide distribution range within 0.5 km–2 km, and the frequency of 40 sr–50 sr was the highest (24.6%), which was similar to the range of 0.5 km–5 km. Large LR (> 60 sr) were mainly distributed in the range of 0.5 km–2.0 km, suggesting that the aerosol in this height range had strongly absorbing ability. Although there were a few large LR (> 60 sr) above 2 km, LR was mainly distributed between 0 sr–40 sr with the highest frequency of 34% between 10 sr and 20 sr.

7. Page 9 line 247: I guess "an effort" should be "order".

Reply: We appreciate the referee's suggestions and have revised it. In order to further research the influences of wind directions on LR and its vertical distribution, cluster analysis of back trajectories was used to study the transport of atmospheric aerosols.

8. Page 12 line 343: change "667-hours" to '667-hour'.

Reply: We have revised it according to referee's suggestions. The aerosol LR at 355nm were retrieved, and the variations of LR and their influencing factors were analyzed in detail based on 667-hour data.

9. Figure 5: x-axis of this figure should be marked by date (not hour), so that readers easily understand seasonal variation of LR in Shanghai.

Reply: The x-axis of Figure 5 has been marked by date (Day/Month/Year) according to referee's suggestions.

10. Figure 6: it is well known that dust aerosols usually show large depolarization

ratio (DR). As descried by the authors, LR of dust is 40-60 sr. However, LR which corresponds to large DR are in range of 100-120 sr. Please explain the reason.

Reply: The referee's comments are very valuable. We have added some analysis to explain why LR which responses to large DR are in range of 100-120 sr in our manuscript.

It is worth noting that LR which responses to large depolarization ratio are in range of 100 sr–120 sr. Generally, the LR of dust aerosol with large depolarization ratio was between 40 sr and 60 sr (Murayama et al., 2004; Noh et al., 2007). Hee et al. (2016) found that 355nm LR of aged forest fire aerosols was relatively large, ranging from 80sr to120sr. And previous studies have found that some aged forest fire aerosols also show large depolarization ratios (Hu et al., 2019; Murayama et al., 2004). There might be two reasons for this phenomenon. One is that dust aerosols on the surface are lifted into the biomass burning plume (Müller et al., 2007), and the other is nonsphericity of particles due to coagulation of smoke particles during aging process (Reid et al., 1998). Therefore, the LR in the range of 100 sr–120 sr corresponds to large DR may be caused by aged forest fire aerosols.

11. Figure 9: Please mark the location of lidar site in all panels.

Reply: The location of lidar site has been marked by blue star in all panels in Figure 9 according to referee's suggestions.

12. The English of manuscript should be further improved before publication.

Reply: We appreciated the referee's suggestions. We reviewed the manuscript carefully and have tried our best to corrected grammatical errors and improve the English of the paper.

[revised manuscript text omitted]

---

## Author Response (AR2)

Dear Editor

Our manuscript acp-2020-1162 entitled "Long term variation of aerosols lidar ratio in Shanghai based on Raman lidar measurement" has been revised according to the editor' comments. We appreciated editor' suggestions and endeavors. In the revised version, we checked and revised the manuscript carefully according to the editor' comments

In the following, we will give an item-by-item response to editor' comments. Editor's comments are in black. Authors' responses are in blue.

Best wishes.
Qianshan He

1. Line 79, "Influencing factor" should be changed to "influence factor".

**R:** We appreciated the editor's suggestions and have revised it. Similarly, we have also changed other same expressions in the manuscript.

[revised manuscript text omitted]